# Modulation of Acute Intestinal Inflammation by Dandelion Polysaccharides: An In-Depth Analysis of Antioxidative, Anti-Inflammatory Effects and Gut Microbiota Regulation

**DOI:** 10.3390/ijms25031429

**Published:** 2024-01-24

**Authors:** Zhu Li, Xinyao Li, Panpan Shi, Pingping Li, Yue Fu, Guifeng Tan, Junjuan Zhou, Jianguo Zeng, Peng Huang

**Affiliations:** 1College of Animal Science and Technology, Hunan Agricultural University, Changsha 410128, China; 2Hunan Key Laboratory of Traditional Chinese Veterinary Medicine, Hunan Agricultural University, Changsha 410128, China; 3College of Veterinary Medicine, Hunan Agricultural University, Changsha 410128, China

**Keywords:** dandelion polysaccharides, intestinal inflammation, gut microbiota, 16S rRNA, LPS

## Abstract

Acute colitis is a complex disease that can lead to dysregulation of the gut flora, inducing more complex parenteral diseases. Dandelion polysaccharides (DPSs) may have potential preventive and therapeutic effects on enteritis. In this study, LPS was used to induce enteritis and VC was used as a positive drug control to explore the preventive and therapeutic effects of DPS on enteritis. The results showed that DPS could repair the intestinal barrier, down-regulate the expression of TNF-α, IL-6, IL-1β, and other pro-inflammatory factors, up-regulate the expression of IL-22 anti-inflammatory factor, improve the antioxidant capacity of the body, and improve the structure of intestinal flora. It is proved that DPS can effectively prevent and treat LPS-induced acute enteritis and play a positive role in promoting intestinal health.

## 1. Introduction

Acute intestinal inflammation constitutes a critical biological response, initiated by the host organism to counteract deleterious stimuli such as microbial invasion. It has been reported that the duodenum and its microbes have become one of the major factors in various potential metabolic and immune diseases [1,2]. Lipopolysaccharides (LPSs), derived from the cell walls of Gram-negative bacteria, are a predominant factor precipitating intestinal inflammation [3]. The presence of LPSs can substantially disrupt the gut’s microbial homeostasis and exacerbate acute inflammatory conditions. Dysregulation of inflammation, especially when chronic in the gut, is increasingly correlated with a range of pathophysiological conditions. Often, these chronic states have their genesis in acute episodes of inflammation, frequently triggered by LPSs or similar microbial agents. Therefore, the effective management of acute intestinal inflammation, particularly when induced by LPSs, is imperative. Such management not only provides relief from immediate gastrointestinal symptoms but also impedes the progression to chronic inflammatory states, potentially carrying significant broader health implications.

The relationship between inflammation and oxidative stress is complex and well-studied [4,5]. Reactive oxygen species (ROS), generated during inflammation, can further exacerbate tissue damage and the overall inflammatory process [6]. Therefore, antioxidants not only neutralize ROS but also offer the potential to modulate inflammatory pathways. Given the mounting evidence that underscores the critical roles of both inflammation and gut microbiota in health and disease, the quest for safe and effective natural anti-inflammatory agents has witnessed a significant paradigm shift [7,8]. The focus is increasingly on compounds that offer dual benefits: modulating the inflammatory response and positively influencing gut microbiota.

Dandelion (*Taraxacum officinale*) has a long-standing reputation in various traditional medicine systems, such as traditional Chinese medicine, Ayurveda, and multiple forms of Western herbal medicine [9]. Known for its antioxidative and anti-inflammatory properties, it has been used to treat a broad range of conditions, including liver diseases and digestive disorders [10]. In recent years, the focus has shifted towards the plant’s polysaccharides [11], which have shown promising biological activities and low toxicity profiles. This development has catapulted them into the realm of intense scientific research, aimed at substantiating their applicability in modern medicine. Although there is an accumulating body of evidence in favor of the therapeutic activities of dandelion polysaccharides, there is still a gap in our understanding of the mechanisms underlying their anti-inflammatory properties, especially concerning the gut environment. The gut ecosystem is intricately complex, consisting of a finely balanced interaction among microbiota, immune cells, and epithelial barriers [12]. Modulating this environment through interventions like DPSs could have extensive repercussions, affecting not only gut health but also systemic inflammation and immune responses.

In the present study, we aim to explore the anti-inflammatory effects of DPSs using a mouse model. Initially, the mice underwent a one-week pre-feeding period, followed by a two-week regimen of intragastric drug administration. Subsequently, one day after the completion of intragastric administration, the mice were intraperitoneally injected with an LPS. The model is designed to mimic acute gut inflammation and assess the impact of DPS on the inflammatory response. To evaluate the anti-inflammatory and antioxidative effects of the treatment, we used intestinal tissue sectioning and fluorescence quantification techniques to measure gut antioxidative and anti-inflammatory factors, which are critical in the inflammatory process. Additionally, we plan to conduct a comprehensive analysis of the gut microbiota using techniques such as 16S rRNA sequencing. This will help us understand how DPSs modulate the gut microbial community, thereby exerting their anti-inflammatory effects. Using this multifaceted experimental approach, this study aims to provide a comprehensive understanding of the potential benefits of DPSs in anti-inflammatory, antioxidative, and gut microbiota regulation, thereby establishing its scientific basis as a potential therapeutic option.

## 2. Results

### 2.1. Analysis of DPS

#### 2.1.1. The Total Sugar Content of DPS

The standard curve equation is y = 0.3027x + 0.1396, R2 = 0.9982. The total sugar content of DPS was detected to be 9.27% (Table 1).

#### 2.1.2. Molecular Weight of DPS

Calibration curves for lgMp-RT (peak molecular weight), lgMw-RT (weight-average molecular weight), and lgMn-RT (number-average molecular weight) were obtained.

The equation for the lgMp-RT calibration curve is:y = 0.1785x + 11.416, R2 = 0.9986 (1)

The equation for the lgMw-RT calibration curve is:y = 0.1766x + 11.348, R2 = 0.9981(2)

The equation for the lgMn-RT calibration curve is:y = 0.1772x + 11.357, R2 = 0.9988(3)

Based on the standard curve, the calculation formula is obtained to calculate the molecular weight of each sample. The molecular weight spectrum of the sample is shown below, and the calculation results are shown in Table 2.

### 2.2. Effect of DPS on Intestinal Segment Morphology in Mice

Six hours post intraperitoneal LPS injection, aiming at establishing a model of acute enteritis, the mice were dissected, and their duodenum was stained and sliced, as depicted in Figure 1A. Observations at 20× magnification using Case Viewer software 2.4.0 revealed that in the control group, epithelial cells were intact and orderly, with partially disrupted villi, potentially attributable to experimental handling. Conversely, in the model group, epithelial cells were disordered, with fewer goblet cells and irregularly broken villi compared with the control. The positive control group exhibited disorganized epithelial cells and pronounced infiltration of blood cells. In the low-dose group, there was a reduction in blood cell infiltration, an increase in goblet cells, orderly epithelial cells, and elongated villi. The villi in the medium-dose group appeared more intact than in the low-dose group, with an increased number of goblet cells but also heightened blood cell infiltration. In the high-dose group, villi elongation was observed alongside neatly arranged epithelial cells, but with substantial blood cell infiltration and a decrease in goblet cells.

The calculated V/C value is shown in Figure 1B. The height of the bar chart represents the calculated mean value of V/C for each group, the error line is the standard deviation of data for each group, and “*” represents the significant difference (*p* < 0.05). It can be seen that the V/C value of the model group is significantly lower than that of the control group. The V/C value of the DPS groups had a certain degree of recovery, and the effect was better than that of vitamin C. This not only confirms the success of our experiment but also suggests that DPS is more effective than vitamin C in restoring V/C value, which decreases due to aggravation of enteritis. The average expression levels of Claudin-1 and ZO-1 were also found to be upregulated, as illustrated in Figure 1C,D. The comparison between the expression level of ZO-1 in the DPS LD group and the M group showed that DPS had a good effect on promoting the expression of this cell compact linking protein.

### 2.3. Anti-Inflammatory and Antioxidant Effects of DPS on Mice

In the model group, there was a notable increase in the medial expression levels of inflammatory factors such as TNF-α and IL-1β compared with the control group (Figure 2A–D). In contrast, the expression levels in the positive control group and the groups administered varying doses of DPS exhibited a decrease. Notably, the medium-dose DPS group (MD) demonstrated the most significant reduction in TNF-α expression relative to the model group, corroborating the successful establishment of our intestinal inflammation model. Additionally, our findings revealed an elevated expression level of IL-22, a factor instrumental in mitigating inflammation, in the DPS group compared with the positive control group. This was particularly pronounced in the low-dose group versus the model group. At the same time, the medial expression of CAT, Nrf2, HO-1, and NQO1, which are substances related to antioxidant ability, in the DPS groups was closer to the level of the control group than that of the model group (Figure 2E–H). This phenomenon can be seen that DPS can affect the body’s antioxidant capacity.

### 2.4. DPS Regulates Intestinal Flora

Our subsequent analysis focused on the impact of DPS on the intestinal microbiota of mice. Figure 3A illustrates a distinct segregation among the groups, with the constituents of the inflammatory model group exhibiting greater dispersion compared with the others. The Shannon Index is one of the indexes used to measure the species diversity in an ecosystem. The higher the value of the Shannon index, the richer the species diversity in the ecosystem. The data presented in Figure 3B indicate that the species diversity in each treatment group was comparable to that of the control group, particularly noticeable in the low-dose group. Figure 3C,D reveals a distinct microbial composition in each group. At the family level, the main bacteria in each group were Muribaculaceae, Lachnospiraceae, Bacteroidaceae, Lactobacillaceae, and Staphy lococcaceae. At the genus level, the main bacteria that accounted for the majority of the samples in each group were norank_f__Muribaculaceae, Bacteroides, Lactobacillus, Staphylococcus, and Helicobacter. At the family level, the Staphylococcaceae population in the low-dose group was significantly higher than in the other groups. This trend was similarly reflected at the genus level with Staphylococcus. The differences in microbial structure between the different groups demonstrate that DPS affects the gut microbiota. As a biological information data visualization tool, the unique and intuitive layout of the Circos plot provides researchers with an effective way to gain insight into the microbial community structure (Figure 3E). The left side of the outer ring is grouped, and the top 20 abundant Operational Taxonomic Units (OTUs) across each category are on the right. The relationship between the proportion of bacteria in each group is represented by the line in the inner circle, which shows the proportion of bacteria in each group as well as the proportion of bacteria in different groups. The most obvious is Staphylococcaceae, which has a higher percentage in the Low_dose (yellow) group than in other groups. Additionally, Figure 3F delineates the hierarchical interconnections among the primary taxa within a sample community, ranging from the phylum to the genus level, with the size of the nodes indicating the average relative abundance of each taxa.

### 2.5. Effects of DPS on SCFAs in the Intestinal Contents of Mice

The acetic acid, butyric acid, isovaleric acid, and propionic acid contents measured in mouse intestinal contents are shown in Figure 4A–D. The levels of acetic acid, butyric acid, isovaleric acid, and propionic acid were different in each group. Compared with the control group, acetic acid, butyric acid, isovaleric acid, and propionic acid were increased in the low-dose DPS group, and the increase rate was higher than that in other groups. To sum up, the mechanism is shown in the drawing in Figure 5.

## 3. Discussion

Many studies have shown that natural plant polysaccharides can enhance immunity, improve antioxidant capacity, and regulate the structure of intestinal flora. Polysaccharides from Rhododendron principis significantly decreased the levels of TNF-α and IL-6 in both serum and blood and bronchoalveolar lavage fluid in LPS-induced acute lung injury mice by intragastric administration [13]. Ganoderma polysaccharides can inhibit inflammation by regulating gut flora and immune cell function [14]. *Lycium barbarum* polysaccharide can alleviate LPS-induced inflammation by expressing pro-inflammatory factors in immune cells [15]. Our findings highlight the beneficial role of DPS, demonstrating its efficacy in promoting intestinal protection and anti-inflammatory antioxidants. DPS not only alleviates LPS-induced inflammation but also has a good effect on regulating intestinal microbial structure.

Intestinal barrier dysfunction is the pathological manifestation of intestinal inflammation [16]. Intestinal epithelial injury often leads to microbial migration, which in turn causes local immune activation, leading to an abnormal release of pro-inflammatory cytokines, and intestinal epithelial injury makes blood and parenteral tissues more susceptible to bacterial or metabolite infiltration [17]. The ratio of intestinal villus length to crypt depth is often called the villus–crypt ratio, which has important physiological and pathological significance in the evaluation of intestinal mucosal morphological structure. A higher villus–crypt ratio is generally regarded as an indicator of intestinal mucosal health. Meanwhile, goblet cells secrete mucus and help maintain intestinal health [18]. DPS can restore the V/C value that is reduced due to inflammation and increase the number of goblet cells. In clinical investigations, it has been observed that aberrant expression of tight junction proteins like Claudin-1 and ZO-1 is induced by inflammatory mediators, intestinal bacteria, and cytokines [19,20]. Studies have shown that the protein levels of claudin, occludin, and ZO-1 in mice infected with bacteria are negatively correlated with the mRNA levels of IL-6, IL-1β, and TNF-α [21]. In addition, other studies have found that LPS can down-regulate the expression of ZO-1 by affecting the Akt pathway [20]. DPS not only alleviates the damage of intestinal epithelial cells to a certain extent, but also upregulates the cell connexins Claudin-1 and ZO-1 [22], which play an important role in maintaining tissue structure, forming barriers, and regulating intercellular communication, and also regulates inflammatory stress. It can be seen that DPS can maintain intestinal shape and contribute to intestinal health.

DPS may relieve inflammation by regulating the TNF pathway. The downregulation of IL-1β, which has strong pro-inflammatory activity and can induce a variety of inflammatory diseases [23]; the pluripotent pro-inflammatory cytokine TNF-α [24], which is similar to IL-1β and can cause systemic inflammation; and the pro-inflammatory cytokine IL-6 and upregulated anti-inflammatory cytokine IL-22 all confirmed that DPS has anti-inflammatory effects [25,26]. The adjustment of these factors may also be one of the mechanisms by which DPS upregulates ZO-1. The effect of DPS on these inflammatory factors in the TNF pathway may be one of the reasons why DPS can alleviate inflammation.

At the same time, DPS is not only a natural anti-inflammatory agent but also an antioxidant. CAT, Nrf2, HO-1, and NQO1 are key enzymes or genes involved in anti-oxidation in cells [27,28,29], and their expression levels are often considered to be indicators of cellular response to oxidative stress. Nrf2 activates a series of antioxidant genes including CAT, HO-1, and NQO1, regulates their expression, and then up-regulates the activity of antioxidant protease, clears excess ROS, and alleviates oxidative stress damage [30,31]. Under normal circumstances, the expression levels of these proteins or genes of the antioxidant and cellular defense systems remain moderate. This maintains intracellular redox balance. In our experiment, DPS restored abnormal expression to normal levels to a certain extent, thus alleviating oxidative stress in the body, which is conducive to the health of the body.

It is generally believed that the diversity and stability of the intestinal flora affect the health of the host [32]. LPS-induced enteritis disrupts this homeostasis, and DPS reduces this imbalance. Compared with the LPS enteritis model group, the DPS low-dose group raised the relative abundance of Muribaculaceae and Staphylococcaceae bacteria and lowered the relative abundance of Prevotellaceae, Helicobacteraceae, and Oscillospiraceae bacteria. The high relative abundance of Staphylococcaceae may be due to their greater resistance to the outside world than other bacteria [33]. The acidic environment of the gut is closely related to the microbial metabolite SCFAs [34], which protect the body by inhibiting harmful bacteria that are intolerant to acid. We hypothesized that the addition of DPS affected the nutrient metabolism of these gut microbiota, resulting in structural changes. The concentrations of several SCFAs in the DPS group, especially in the low-dose group, were significantly higher than those in the model group, so the intestinal pH was lower, which inhibited the growth and reproduction of potentially harmful bacteria.

In summary, LPS can induce a variety of diseases through inflammatory pathways, while DPS can affect the expression of inflammatory factors in the TNF signaling pathway and the expression of antioxidant active substances in the body and alleviate inflammation by regulating the composition of intestinal flora. This provides some new references for subsequent relevant research.

## 4. Methods and Materials

### 4.1. Ethics Statement

This experiment was performed according to the guidelines for the care and use of laboratory experimental animals in China. This study was approved by the Biomedical Research Ethics Committee, Hunan Agricultural University (Permit No. HAU2022024 15 March 2022).

### 4.2. Analysis of Polysaccharides

#### 4.2.1. Total Sugar Content Determination

To begin with, the first step is to weigh 8 mg of standard dextran in a 2 mL volumetric flask and fill it up with water until the mark. Then, prepare standard solutions of 8 mg/mL, 4 mg/mL, 2 mg/mL, 1 mg/mL, 0.5 mg/mL, 0.25 mg/mL, and 0.125 mg/mL, each with a volume of 1 mL. After that, take 100 μL of every solution, add 6% phenol 100 μL, and concentrated sulfuric acid 1 mL. Let it sit for 10 min, then shake well, and leave it at room temperature for 20 min before measuring its absorbance at 490 nm. The same procedure should be followed using pure water as a control. A standard curve with polysaccharide concentration on the *x*-axis and absorbance on the *y*-axis should be constructed. After that, accurately weigh 20 mg of the sample dissolved in 200 μL of water and follow the above steps to measure the absorbance for the determination of polysaccharide content using the standard curve.

#### 4.2.2. Molecular Weight Determination

The first step is to prepare the sample and standard solutions (Table 3). Precisely weigh the sample and standard and prepare the sample into a 5 mg/mL solution. Centrifuge at 12,000× *g* for 10 min, filter the supernatant through a 0.22 μm microporous membrane and then transfer the sample into a 1.8 mL vial for injection.

The second step is to perform chromatographic analysis using a BRT105-104-102 series gel column (8 × 300 mm) with a mobile phase of 0.05 M NaCl solution. Set the flow rate at 0.6 mL/min, the column temperature at 40 °C, and the injection volume at 25 μL. For this step, use the RI-10A differential detector.

### 4.3. Animal Experiment and Sample Collection

Forty-eight healthy male ICR mice were systematically allocated into six distinct groups: the control, model, positive control, and low-, medium-, and high-dose DPS groups, each granted unrestricted access to food and water during a seven-day acclimatization phase. Following a 14-day period of consistent gavage, with each administration amounting to 0.3 mL, the control and model groups received distilled water, whereas the positive control group was administered a vitamin C solution (200 mg/kg), which entailed daily gavage at noon. The DPS groups received dosages of 100, 200, and 300 mg/kg for the low-, medium-, and high-dose groups, respectively. Subsequent to the 14-day gavage phase, all groups, barring the control, were intraperitoneally injected with 0.3 mL of LPS, while the control group received an equivalent volume of distilled water. Six hours post-injection, the mice were euthanized, and duodenal and liver samples were harvested, meticulously cataloged, and preserved under optimal conditions for further analysis, as illustrated in Figure 6.

### 4.4. Duodenal Morphological Observations

Following dissection, the duodenal segments from the mice were meticulously excised and preserved in 4% paraformaldehyde for a duration of 24 h. The tissue samples then underwent a series of processing steps including gradient dehydration with ethanol, clarification using xylene, and subsequent embedding in paraffin wax. Thin sections were skillfully prepared using a microtome and were then carefully mounted onto slides for the processes of baking and drying. These slides were processed for dewaxing using a combination of xylene and ethanol (H&E, C0105S, Beyotime, Beijing, China). Once prepared for staining, the tissue sections were treated with hematoxylin (H&E, C0105S, Beyotime, Beijing, China) for five minutes. This was followed by a differentiation phase in a 1% hydrochloric–ethanol solution and a brisk rinse in distilled water. The sections were then stained with an eosin solution for one minute and subsequently rinsed twice with distilled water. Concluding the preparation, the sections were dehydrated, dismounted, air-dried, and sealed with a neutral adhesive for preservation. For comprehensive examination, the samples underwent microscopic imaging to capture detailed visualizations, selecting a better film effect for display.

### 4.5. qPCR Assay and Detection of Antioxidant Activity Factors

The expression of Claudin-1, ZO-1, TNF-α, IL-22, IL-6, IL-1β, CAT, Nrf2, HO-1, and NQO1 in duodenal tissues was assessed using the real-time quantitative (qPCR) method. Duodenal tissues were mixed with RNAiso Plus, homogenized, and total RNA was extracted. The primer sequences are shown in Appendix A. The removal of genomic DNA was performed using 5× gDNA Clean Buffer and RNase-free water. Subsequently, reverse transcription was carried out using Evo M-MLV RT Premix according to the manufacturer’s instructions for qPCR. Subsequently, reverse transcription was carried out using Evo M-MLV Mix kit with gDNA Clean for qPCR Premix according to the instructions. The SYBR@ Green Premix Pro Taq HS qPCR Kit AG11701 was mixed with cDNA, and the reaction was carried out using the Real-Time PCR System (Thermo Fisher Scientific, Waltham, MA, USA) with 1 cycle of the pre-denaturation reaction: 95 °C for 30 s and 40 cycles of the polymerase chain reaction. The mRNA expression levels of target genes were calculated using the 2^−ΔΔCT^ method. Each group detected 6 values for each indicator, and the final average was taken as the indicator detection result.

### 4.6. Fecal Microbiota 16S rRNA Analysis

The intestinal contents of 6 mice in each group were collected, packaged, and numbered, and the samples were sent to Mejorbio (Shanghai, China) for sequencing. The samples were analyzed using Mejorbio’s cloud platform services calculator. The raw data are available in PRJNA1060481.

### 4.7. Determination of Short-Chain Fatty Acids in Intestinal Contents

Six samples were taken from each group, and the final result was averaged. The procedure was commenced by taking 100 milligrams of fecal matter and blending it with a volume of ultrapure water that was double its size, ensuring thorough vortexing under ice bath conditions. Next, this mixture was centrifuged at 12,000× *g* for 10 min at 4 °C, and then the resultant supernatant was collected carefully. An Agilent gas chromatograph (GC) equipped with a silica gel capillary column (DB-Wax, J&W, 30 m × 0.25 mm I.D.) was used, and the collected samples were passed through a silica gel capillary for filtration. In our analysis, we meticulously set the chromatographic conditions, with the column temperature progressively increasing from 50 °C to 220 °C at a rate of 4 °C/min. The temperatures of the injection port and detector were precisely maintained at 225 °C and 250 °C, respectively. Helium served as the carrier gas, with a flow rate maintained at 1.0 mL/min and a split ratio set at 1:20. A volume of 1 µL from each standard and sample was injected and subjected to continuous analysis for a duration of 25 min. The chromatogram peaks are reported to standards, see Appendix A.

### 4.8. Data Analysis and Mapping

Statistical analyses were meticulously conducted utilizing SPSS software (version 21.0) and the R programming language. The results are presented as mean values accompanied by standard deviations (SDs). For comparisons between two distinct groups, the *t*-test was used, whereas analysis of variance (ANOVA) was utilized for scenarios involving multiple groups. A significance threshold was set at *p* < 0.05 to demarcate statistically significant differences. The mechanism diagram was expertly crafted using the Figdraw website.

## 5. Conclusions

DPS exhibits remarkable potential in maintaining intestinal health by enhancing the body’s anti-inflammatory and antioxidant capabilities. In comparison with vitamin C, DPS demonstrates a superior efficacy to a certain extent. Its multifaceted benefits extend beyond conventional supplementation, as DPS also plays a pivotal role in regulating the composition and activity of intestinal microorganisms. Furthermore, while DPS contributes to the modulation of intestinal microflora, an excessive intake might disrupt the delicate balance of gut bacteria, potentially leading to dysbiosis. Research is underway to better understand the nuanced effects of DPS on the gut microbiome and its implications for overall health.

In summary, while DPS holds promise for enhancing intestinal health and bolstering the body’s defense mechanisms, it is crucial to approach its supplementation cautiously. Further work remains to be completed to select the optimal dose.

## Figures and Tables

**Figure 1 ijms-25-01429-f001:**
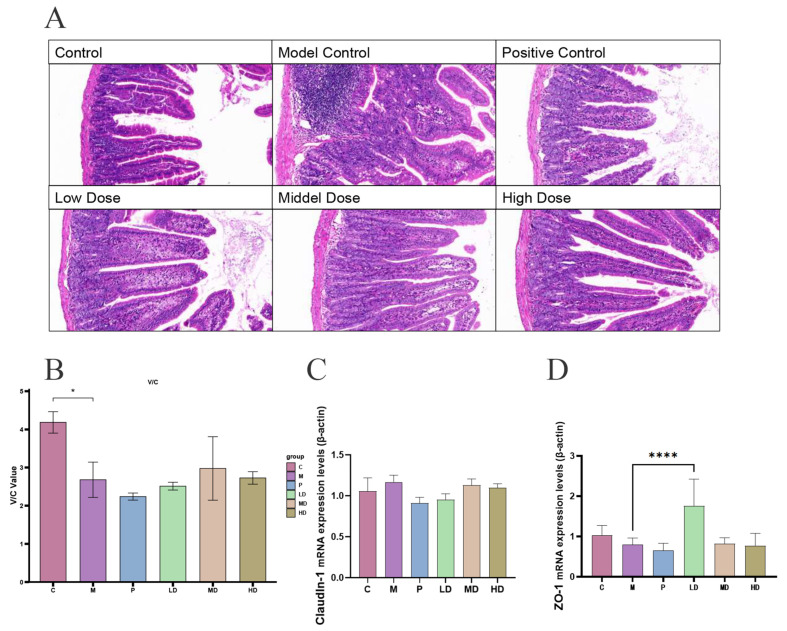
Duodenal section after drug treatment. (**A**) Intestinal section diagram, observations at 20× magnification using Case Viewer software. (**B**) Average V/C value of each group. The horizontal coordinate is the abbreviation of each group name: control (C), model (M), positive control (P), low-dose (LD), medium-dose (MD), and high-dose (HD). The ordinate is the ratio of intestinal villi length to crypt depth (V/C). (**C**,**D**) Cell compact linking protein: Claudin-1 and ZO-1. There are 6 samples in each group, and the final result is averaged. The error bars represent the standard deviation of the data. * means *p* < 0.05, and **** means *p* < 0.0001.

**Figure 2 ijms-25-01429-f002:**
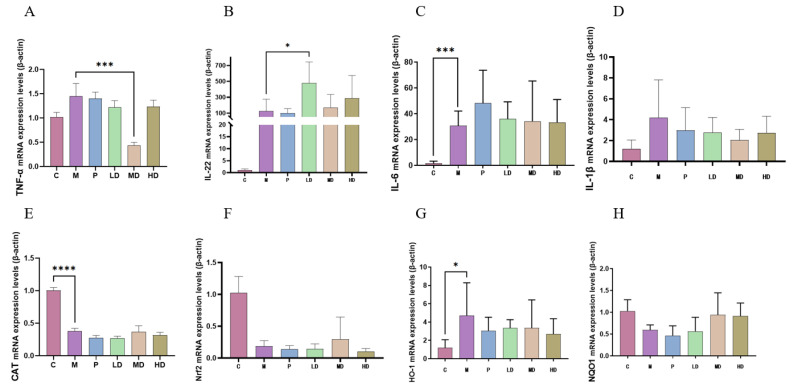
Anti-inflammatory and antioxidant effects of DPS on LPS-induced enteritis model. The average expression value of cytokines involved in anti-inflammatory activity: (**A**–**D**) TNF-α, IL-22, IL-6, and IL-1β. The average expression value of indicators related to antioxidant capacity: (**E**–**H**) CAT, Nrf2, HO-1, and NQO1. There are 6 samples in each group, and the final result is averaged. The error bars represent the standard deviation of the data. * means *p* < 0.05, *** means *p* < 0.001, and **** means *p* < 0.0001.

**Figure 3 ijms-25-01429-f003:**
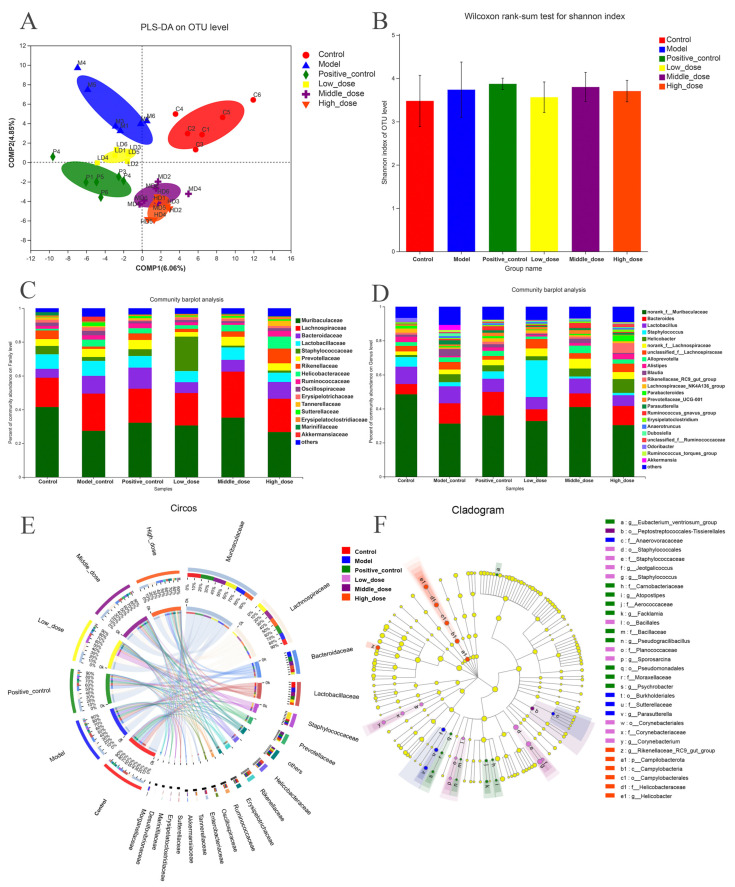
DPS regulates intestinal flora in mice, each group had 6 samples. (**A**) PLS-DA on the OUT level, different groups are distinguished by different colors and shapes. (**B**) The Shannon index of each group, where the error bars represent the standard deviation of the data. (**C**,**D**) Species composition at the family and genus levels, where the horizontal axis is the name of each group, and the vertical axis is the percent of community abundance at the family or genus level. (**E**) Circos plot, where the left side is the group name, the right side is the relative abundance OTU of the top 20 abundance values, the inner ring color corresponds to the OTU or group name, and the outer ring is the abundance of the species or group. (**F**) Cladogram, where the taxonomic branching diagram shows the hierarchical relationships of the major taxa in a sample community from phylum to genus (inner ring to outer ring), and the node size is relative to the average relative abundance of taxa.

**Figure 4 ijms-25-01429-f004:**
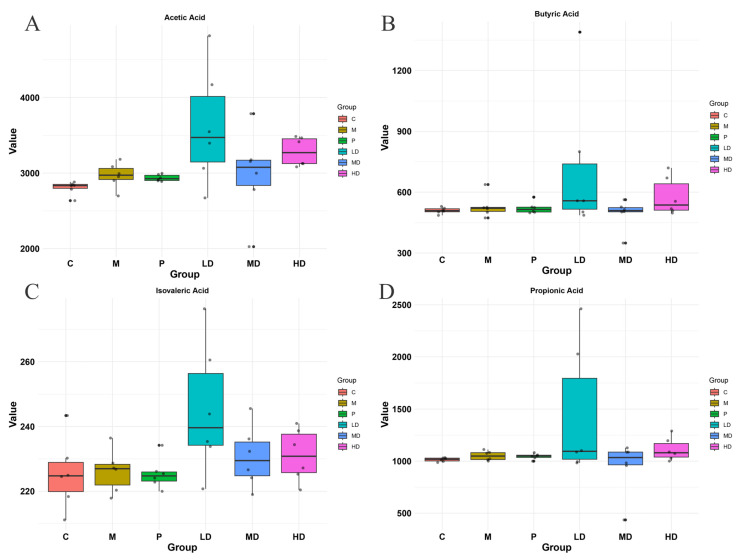
SCFAs were detected in each group. (**A**) Acetic acid, (**B**) butyric acid, (**C**) isovaleric acid, and (**D**) propionic acid content comparison in mouse intestinal contents. The horizontal axis is the group name of each group, and the vertical axis is the value of SCFAs. Each group had 6 observations. The scatter point in the box plot represents the observed value, and the three lines of the box from top to bottom are the top quartile, the mean, and the bottom quartile. The error bars represent the standard deviation of the data.

**Figure 5 ijms-25-01429-f005:**
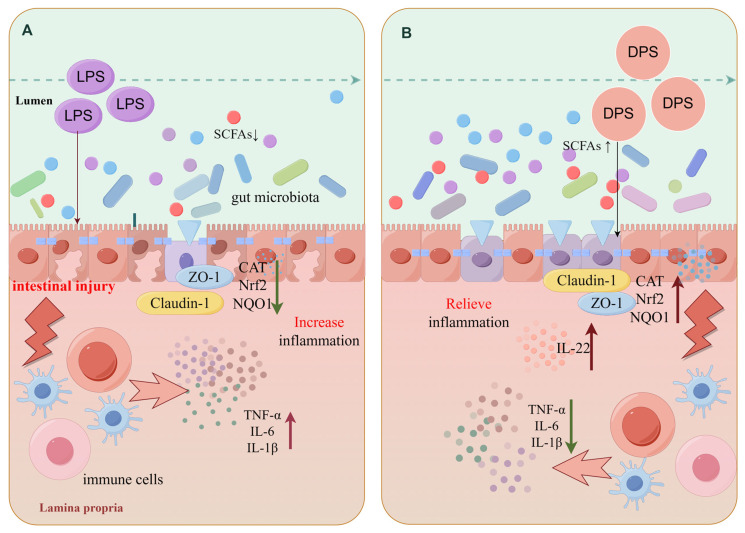
LPS destroys intestinal barrier to induce acute enteritis (**A**), DPS relieves enteritis and protects intestinal tract (**B**). Schematic diagram of DPS alleviating LPS-induced enteritis. LPS induces acute enteritis, destroys the intestinal barrier, activates immune cells to secrete inflammatory activity and antioxidant-related factors, and affects intestinal flora. DPS protects the gut, regulates immunity, and promotes the normal structure of the intestinal state and related factors and intestinal flora.

**Figure 6 ijms-25-01429-f006:**
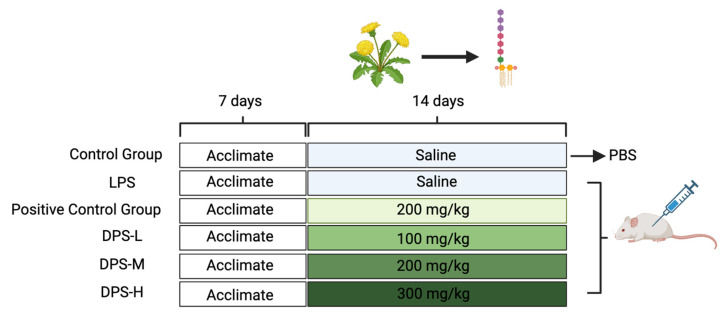
Animal experimental design after pre-feeding. A total of 48 mice were randomly divided into 6 groups with 8 mice in each group.

**Table 1 ijms-25-01429-t001:** Relationship between absorbance and total sugar content of DPS.

	1-1 Absorbance Value	1-2 Absorbance Value	1-3 Absorbance Value	Mean	Sample Size (mg/mL)	Total Sugar Content %
DPS	2.583	3.181	3.075	2.946	100	9.27

**Table 2 ijms-25-01429-t002:** Molecular weight of RIPS.

RT (min)	lgMp	lgMw	lgMn	Mp	Mw	Mn	Peak Area Ratio %
37.523	4.7	4.7	4.7	52,257	52,655	51,042	51.357
42.08	3.9	3.9	3.9	8030	8254	7951	48.643

**Table 3 ijms-25-01429-t003:** Standards.

Standards	Manufacturers	Lot Number	Storage Condition	Purity	Expiry Date
Dextranstandards 1152	Yuanye Bio-Technology (Shanghai China)	A16A8L41850	Sealed storage	99%	Two years
5000	Sigma (Saint Louis, MO, USA)	102084138	Sealed storage	≥99%	Two years
11600	Sigma	102136543	Sealed storage	99%	Two years
23800	Sigma	102124529	Sealed storage	≥97%	Two years
48600	Sigma	102104509	Sealed storage	≥99%	Two years
80900	Sigma	102108375	Sealed storage	>98%	Two years
148000	Sigma	102089360	Sealed storage	>98%	Two years
273000	Sigma	102110878	Sealed storage	98%	Two years
409800	Sigma	102124507	Sealed storage	>98%	Two years
667800	Sigma	102104510	Sealed storage	>98%	Two years

The unit of molecular weight is Da, and the full name is Dalton.

## Data Availability

Data availability in PRJNA1060481.

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
