# Peer review of "Modulation of Acute Intestinal Inflammation by Dandelion Polysaccharides: An In-Depth Analysis of Antioxidative, Anti-Inflammatory Effects and Gut Microbiota Regulation"

_ijms, 2024, doi:10.3390/ijms25031429_

Round 1

Reviewer 1 Report

Comments and Suggestions for Authors

The manuscript suggests that DPS may have potential preventative and therapeutic effects on enteritis. The authors demonstrated that DPS can effectively prevent and treat LPS-induced acute enteritis and promote intestinal health. However, before publication, the following points need to be addressed:

1.      The figures need improvement, such as providing clearer figures and explaining the significance in all graphs. Additionally, the meaning of the error bars should be clarified.

2.      The molecular mechanism behind the upregulation of ZO-1 mRNA and the cytokines needs further explanation.

3.      The results in Figure 3 should be presented more clearly and in a more focused manner

4.      The authors should discuss the impact of DPS on the antiviral effects in the gut. Please refer to PMID 32555175 and 33839152 for further information.

Author Response

Dear reviewer,

Thank you for your detailed review of our paper (ijms-2833956) and your valuable comments. Your suggestions are greatly appreciated, and here is our response to each point:

1.Question:The figures need improvement, such as providing clearer figures and explaining the significance in all graphs. Additionally, the meaning of the error bars should be clarified.

Answer:Thank you for your valuable feedback. We have revised the figures in the manuscript to enhance clarity and ensure they effectively communicate the data. Each graph now includes a detailed explanation of its significance. Additionally, we have clarified the meaning of the error bars to better represent the data variability. We appreciate your guidance in improving the quality of our work and hope that these revisions meet the required standards.(Please see lines 124-126, 131-133,178 and 198-199,red)

2.Question:The molecular mechanism behind the upregulation of ZO-1 mRNA and the cytokines needs further explanation.

Answer:Thank you for your constructive comment. In response to your request for a more detailed explanation of the molecular mechanisms behind the upregulation of ZO-1 mRNA and the cytokines, we have expanded this section in our manuscript. We now provide a more comprehensive discussion of the potential pathways and interactions involved, supported by current literature and theoretical models. We believe these additions will significantly enhance the understanding of the mechanisms at play in our study.(Please see lines 227-236, 243-245 red)

3.Question:The results in Figure 3 should be presented more clearly and in a more focused manner

Answer:Thank you for your feedback regarding Figure 3. We have revised the figure to ensure that the presentation of the results is clearer and more focused. This revision includes refining the graphical representation and enhancing the annotations for better clarity and comprehension. We hope that these improvements adequately address your concerns and make the data more accessible and understandable to the readers.(Please see Figure.3 and lines 153-154, 156-171)

4.Question:The authors should discuss the impact of DPS on the antiviral effects in the gut. Please refer to PMID 32555175 and 33839152 for further information.

Answer:Thank you for your suggestion to consider the impact of DPS on the antiviral effects in the gut and for referencing the articles (PMID 32555175 and 33839152). We acknowledge the importance of these studies; however, it is important to note that the focus and context of our research differ significantly from these references. Our study Our research does not involve antiviral, making a direct comparison or discussion in relation to these articles less relevant. Nonetheless, we appreciate your input and will consider broader literature where applicable to provide context to our findings.

We are deeply honored by your guidance and believe that our improvements will further improve the quality of the paper. Thank you again for your patience and professional review.

Sincere thanks

Reviewer 2 Report

Comments and Suggestions for Authors

The manuscript titled “Modulation of acute intestinal inflammation by dandelion 2 polysaccharides: an in-depth analysis of antioxidative, anti-3 inflammatory effects and gut microbiota regulation” investigated the effect of dandelion 2 polysaccharides on acute intestinal inflammation. The authors have given some interesting results and some revisions should be modified as follows:

1.      Why choose the duodenum instead of the colon to verify the regulatory effect of dandelion polysaccharides on acute enteritis?

2.      Please appropriately highlight the phenomenon described by the H&E results in Figure 1A for better understanding by readers, and point out the scale or magnification of the image.

3.      Please add the vertical axis title in Figure 1B, and Figure 4.

4.      Please add the number of animal samples from each group in the graphic annotations, as well as the level of significance represented by different numbers of “*”.

5.      Please distinguish the symbols that indicate significant differences between the model group and the control group, as well as between the medication group and the model group.

6.      In the section of discussion, parts involved in Nrf2/HO-1 pathway, authors can refer to https://doi.org/10.3390/md21060360. Parts involved in oxidative stress and inflammation can refer to https://doi.org/10.1016/j.scitotenv.2023.164808.

7.      The conclusion was too short, and the defects of this paper or further studies can be reflected in conclusion.

Author Response

Dear reviewer,

Thank you for your detailed review of our paper (ijms-2833956) and your valuable comments. Your suggestions are greatly appreciated, and here is our response to each point:

1.Question:Why choose the duodenum instead of the colon to verify the regulatory effect of dandelion polysaccharides on acute enteritis?

Answer:Thank you for your question regarding our choice of the duodenum over the colon for studying the regulatory effects of dandelion polysaccharides on acute enteritis. As mentioned in the introduction of our manuscript (lines 26-28, red), this decision is thoroughly explained. We have provided a detailed rationale based on specific scientific considerations that align with the objectives and scope of our study.

2.Question:Please appropriately highlight the phenomenon described by the H&E results in Figure 1A for better understanding by readers, and point out the scale or magnification of the image.

Answer:Thank you for your inquiry regarding the enhancement of Figure 1A. We have updated the figure caption to better highlight the phenomena observed in the H&E results, as per your suggestion. This includes a clearer description of the key features and specifying the scale or magnification of the image. Additionally, the corresponding text in the document has been modified and highlighted for easier reference (as detailed in the Figure 1 annotation, document lines 97-99 and 120-126, red). We hope these revisions make the information more accessible and comprehensible to our readers.

3.Question:Please add the vertical axis title in Figure 1B, and Figure 4.

Answer:Thank you for your query. We have made the necessary modifications to Figure 1.B and Figure 4, including the addition of vertical axis titles. These updates aim to enhance the clarity and comprehensibility of the figures. We believe these revisions will facilitate a better understanding of the data presented.

4.Question:Please add the number of animal samples from each group in the graphic annotations, as well as the level of significance represented by different numbers of “*”.

Answer:Thank you for your suggestion. We have updated the graphic annotations in the figures to include the number of animal samples from each group (Please see lines 120-126,131-133, 176, 196, 306-307, 351-352 354, and 358, red). Additionally, we have also clarified the level of significance represented by the different numbers of asterisks. These modifications have been highlighted in the document for easy identification. We hope these changes adequately address your request and enhance the clarity of our data presentation.

5.Question:Please distinguish the symbols that indicate significant differences between the model group and the control group, as well as between the medication group and the model group.

Answer:Thank you for your feedback. We have made revisions in the figures and corresponding sections of the manuscript to distinguish the symbols indicating significant differences. These include differentiating between the model group and control group, as well as between the medication group and the model group. The changes have been highlighted in red for clearer visibility and ease of understanding. We hope that these adjustments effectively address your query and improve the clarity of our results.

6.Question:In the section of discussion, parts involved in Nrf2/HO-1 pathway, authors can refer to https://doi.org/10.3390/md21060360. Parts involved in oxidative stress and inflammation can refer to https://doi.org/10.1016/j.scitotenv.2023.164808.

Answer:Thank you for pointing out the relevant references. We have incorporated these sources into our discussion section, particularly focusing on the Nrf2/HO-1 pathway and the aspects of oxidative stress and inflammation. The modifications and additions have been made in lines 249-251 of our document and are highlighted in red for clarity. We believe these references provide valuable support and context to our discussion.

7.Question:The conclusion was too short, and the defects of this paper or further studies can be reflected in conclusion.

Answer:We have revised and expanded the conclusion section, now encompassing lines 380-391 of our document, to provide a more comprehensive summary of our findings and their implications. The revised conclusion also includes a discussion of the limitations of our study and potential directions for future research, as highlighted in yellow for easy identification. We hope these revisions meet your expectations and effectively conclude the paper.

We are deeply honored by your guidance and believe that our improvements will further improve the quality of the paper. Thank you again for your patience and professional review.

Sincere thanks

Reviewer 3 Report

Comments and Suggestions for Authors

Your paper, Modulation of Acute Intestinal Inflammation by Dandelion Polysaccharides: An In-Depth Analysis of Antioxidative, Anti-inflammatory Effects and Gut Microbiota Regulation presents interesting experimental observations developed in discussion section, but I observed deficient in development of laboratory methods.

To improve the quality of the manuscript I will suggest major revision and resubmission:

1.  About your methods, you must present in tables or figures the media for indicators which you determine. Also, for the results, you must calculate the media.

2.  In material and methods section, you didn’t describe the inflammatory and antioxidant methods - TNF-α, IL-22, IL-6, IL-1β; CAT, Nrf2, HO-1, NQO1.

3. In chapter -4.7. Determination of short-chain fatty acids in intestinal contents by gas-chromatography, please attach peaks of chromatogram reported to standards.

4- Please revise introduction, discussion, and conclusion sections to be in correlation with the material and methods.

Author Response

Dear reviewer,

Thank you for your detailed review of our paper (ijms-2833956) and your valuable comments. Your suggestions are greatly appreciated, and here is our response to each point:

1.Question:About your methods, you must present in tables or figures the media for indicators which you determine. Also, for the results, you must calculate the media.

Answer: We have taken your comments into account, so we have determined in the manuscript that all calculations and results are based on medial and have revised the corresponding parts, please see lines 108-109,115, 124-126, 131-133, 135, 144, 198, 351-352, and 358.

2.Question:In material and methods section, you didn’t describe the inflammatory and antioxidant methods - TNF-α, IL-22, IL-6, IL-1β; CAT, Nrf2, HO-1, NQO1.

Answer:Thank you for your comment. We realize the oversight in not detailing the inflammatory and antioxidant methods used to measure TNF-α, IL-22, IL-6, IL-1β; CAT, Nrf2, HO-1, and NQO1 in the materials and methods section. We will rectify this by providing a comprehensive description of these methods in the revised manuscript (Please see lines 339-352). This will include the specific protocols and assays used for each biomarker, ensuring a clear understanding of our methodology. We appreciate your attention to detail and the opportunity to improve the completeness of our paper.

3.Question:In chapter -4.7. Determination of short-chain fatty acids in intestinal contents by gas-chromatography, please attach peaks of chromatogram reported to standards.

Answer:Thank you for your suggestion regarding the addition of chromatogram peaks for standards in the gas-chromatography section of our paper. We have included this information in the supplementary tables to provide a clearer understanding of the chromatographic profiles of the short-chain fatty acids measured in intestinal contents. We hope this addition will enhance the clarity and comprehensibility of our analytical methods.(line 370,red)

4.Question:Please revise introduction, discussion, and conclusion sections to be in correlation with the material and methods.

Answer:Thank you for informing us about the revisions made to the introduction, discussion, and conclusion sections of our manuscript. This coherence is essential for the overall consistency and clarity of the research paper. We appreciate your attention to these important aspects of our manuscript.

We are deeply honored by your guidance and believe that our improvements will further improve the quality of the paper. Thank you again for your patience and professional review.

Sincere thanks

Round 2

Reviewer 1 Report

Comments and Suggestions for Authors   The author has answered my reviews.

Reviewer 2 Report

Comments and Suggestions for Authors

It can be accepted in the present form.

Reviewer 3 Report

Comments and Suggestions for Authors

Accept in present form.